# Application of machine learning methods for predicting childhood anaemia: Analysis of Ethiopian Demographic Health Survey of 2016

Solomon Hailemariam Tesfaye *, Binyam Tariku Seboka , Daniel Sisay

School of Public Health, College of Health Sciences and Medicine, Dilla University, Dilla, Ethiopia

* solomon0917242124@gmail.com

**Data Availability Statement:** The data underlying the results presented in the study are available from DHS programme, https://dhsprogram.com/data/available-datasets.cfm.

## Abstract

Childhood anaemia is a public health problem in Ethiopia. Machine learning (ML) is a growing in medicine field to predict diseases. Diagnosis of childhood anaemia is resource intensive. The aim of this study is to apply machine learning (ML) algorithm to predict childhood anaemia using socio-demographic, economic, and maternal and child related variables. The study used data from 2016 Ethiopian demographic health survey (EDHS). We used Python software version 3.11 to apply and test ML algorithms through logistic regression, Random Forest (RF), Decision Tree, and K-Nearest Neighbours (KNN). We evaluated the performance of each of the ML algorithms using discrimination and calibration parameters. The predictive performance of the algorithms was between 60% and 66%. The logistic regression model was the best predictive model of ML with accuracy (66%), sensitivity (82%), specificity (42%), and AUC (69%), followed by RF with accuracy (64%), sensitivity (79%), specificity (42%), and AUC (63%). The logistic regression and the RF models of ML showed poorest family, child age category between 6 and 23 months, uneducated mother, unemployed mother, and stunting as high importance predictors of childhood anaemia. Applying logistic regression and RF models of ML can detect combinations of predictors of childhood anaemia that can be used in primary health care professionals.

## Introduction

Anaemia is a deficiency of haemoglobin in the blood, which can be caused either by too few red blood cells or too little haemoglobin in the cells [1]. The causes of anaemia are multidimensional [2]. Such causes include haemorrhage, bone marrow dysfunction [1], nutrient deficiencies such as vitamins A, *riboflavin*, *pyridoxine*, *cobalamin*, C, D, and E, folate and copper, diseases such as malaria, Tuberculosis, HIV and parasitic infections [2, 3]. In children under five years of age, iron deficiency is the most common cause of anaemia [3, 4]. A meta-analysis of studies from countries with low, medium, and high human development indexes shows that 50% of anaemia is attributable to iron deficiency [5]. Anaemia results in poor cognitive development and motor development outcomes in children [2]. Globally iron deficiency anaemia is common among children younger than five years [6, 7].

**Funding:** The authors received no specific funding for this work.

**Competing interests:** The authors have declared that no competing interests exist.

Several studies have shown that anaemia is a public health problem in Ethiopia [8–15]. The prevalence from these studies ranges from12% to 59% in children younger than five years. The Ethiopian government set target to reduce the prevalence of anaemia in children age younger than 5 years from 39% to 24% in 2020 [16]. However, according to findings from studies mentioned above Ethiopia left far behind to attaining the target. Identifying and treating anaemia is one of the strategies to achieve the target.

In children between 6 and 59 months of age, anaemia can diagnosed when the concentration of haemoglobin falls below 11 g/dl at sea level [17]. Clinical signs, like palmar pallor, can be used to assess anaemia [2]. However, in resource-poor settings, haematological assessments are not possible. Moreover, palmar pallor assessment of child is challenging for primary health care professionals. Therefore, it is very important to evaluate the ability of predictors of anaemia, which can easily be identified by mid-level health workers in primary health care unit to predict anaemia.

Machine learning (ML) is a technology that enables machines to learn from data and experience and to make decisions and predictions [18]. ML application is becoming increasing in medical fields. Diagnoses of disease and outcome prediction are the two areas that benefit from the application of ML [19]. ML is working based on algorithms sets of mathematical procedures which describe the relationship between variables and develop diagnosis [19, 20]. ML algorithms focus on making predictions as accurate as possible, while traditional statistical methods are aimed at inferring relationship between variables [21]. Unlike traditional statistical methods machine learning algorithms are data driven and free from prior assumptions [22]. ML shows improvement over traditional methods in predicting diseases in health care fields [21, 22].

The diagnosis of childhood anaemia is resource intensive and challenging particularly in rural settings where there is scarcity of resources [21]. Previous studies show improved ML performance in predicting childhood anaemia [23, 24]. Therefore, the aim of this study is to apply a machine learning (ML) algorithm to predict childhood anaemia using the already established risk factors for childhood anaemia. We believe that the findings from this analysis will help healthcare providers to easily identify and treat/refer children with anaemia (Box 1).

### Box 1. Current evidence on childhood anaemia and contribution of the paper

#### Evidence on the subject

Childhood anaemia is a public health problem in Ethiopia. The Ethiopian government implemented strategies to decrease childhood anaemia. Diagnosing childhood anaemia requires technologies not accessible for mid-level health care workers in rural health facilities. Previous studies in Ethiopia focus on risk factors of childhood anaemia.

#### What this study adds

We report socio-demographic, economic, child and maternal health related predictors of childhood anaemia that can be applied by mid-level health workers to diagnose childhood anaemia at health facilities where there is resource scarcity to diagnose anaemia.

## Materials and methods

### Ethics statement

The present analysis uses data from EDHS approved by the national research ethics and review committee. Permission to analyse data; granted by the Demographic and Health Survey (DHS) program. The EDHS data used in this analysis was accessed from the public domain through the DHS website (www.dhsprogram.com). The investigators had requested permission to use the data set on the DHS website to analyse. We accessed data on July 12, 2023. The authors didn't have access to identify individual participants. According to the Demographic and Health surveys programme written consent was obtained from guardians prior to data collection.

### Dataset and study population

This study is based data from 2016 [25] Ethiopian Demographic and Health Surveys (EDHS), a nationally representative household survey. The survey conducted every five years collect data on a wide range of population, nutritional status of women and child, and women and child health. The survey was conducted by the Central Statistical Agency (CSA) in collaboration with Federal Ministry of Health (FMoH) with technical assistance from ICF international and funded by USAID. The survey used a two-stage stratified cluster sampling procedures. Clusters were selected at first stage. In the second stages of selection, households were selected. Haemoglobin was measured from children between 6 and 59 months of age from the selected household. Cases with multiple missing values were excluded from analysis. Analysis is based on cases with complete information. The sample size used for this analysis was based on a weighted sample of 8482 children age 6–59 months for descriptive results.

### Study variables and measurements

For this study, the outcome variable is childhood anaemia. Blood specimen for anaemia testing was collected from children aged 6–59 months. Consent was obtained from their parents or adults responsible for them. For children aged 12–59 months blood samples were taken from a finger prick and heel prick for children aged 6–11 months. The blood samples were collected in a microcuvette. Haemoglobin analysis was carried out using a battery-operated portable HemoCue analyser and results were reported on-site. For the current analysis anaemia was measured as a binary outcome variable. Thus children having anaemia were coded as 1 and not having anaemia were coded as 0. Children were considered anaemic if their altitude-adjusted haemoglobin level was below 11 g/dl [17].

Thirteen predictor variables (features) used in this analysis; were selected based on their association with childhood anaemia from previous studies done both in Ethiopia and elsewhere [6, 10, 12, 15, 26–29]. These are place of residence ("urban", "rural"), mother's education levels ("no education", "primary", "secondary", "higher"), mother's age ("<20", "20–29", "30–39", "> = 40"), maternal anaemia ("yes", "no"), employment status of mother ("employed", "unemployed"), child gender ("male", "female"), child age in month ("6–23", "24–9"), childhood morbidity (if child has fever or diarrhoea or acute respiratory infection ["yes", "no"]), drug for parasites within 6 months ("yes", "no"), child stunting status ("yes", "no"), child wasting status ("yes", "no"), household wealth quintiles ("poorest", "poorer", "middle", "richer", "richest"), household water source ("improved", "non-improved").

During the survey data was collected from households ranging from a television to a bicycle or car possession, in addition to housing characteristics such as source of drinking water, toilet

facilities, and flooring materials. Households are given scores based on the number and kinds of consumer goods they own. These scores are derived using principal component analysis. National wealth quintiles are compiled by assigning the household score to each household [25].

## Statistical analysis

Data weighting, cleaning, and descriptive statistics was carried out using STATA version 15. Data were weighted using sampling weight, primary sampling unit, and strata to restore the representativeness of the data and to get more accurate results. The percentage was used to report results.

We used Python software version 3.11 to apply machine learning (ML) methods for predicting childhood anaemia. We use the four commonly used ML algorithms [18], such as; logistic regression (LR), random forest (RF), K-nearest neighbours (KNN), and decision tree (DT). Logistic regression is used to analyse binary data and estimate the probability that an instance belongs to a particular class. Random forest involves selecting bootstrap variables from the training dataset and fitting a decision tree on each. The k-nearest neighbour, or KNN, algorithm is another nonlinear machine learning algorithm applied to both classification and prediction problems that predict a class label directly. It is a simple algorithm that stores all available cases and classifies any new cases by taking a majority vote of its k neighbours. The case is then assigned to the class with which it has the most in common. The decision tree algorithm is one of the popular algorithms used for classifying problems. It works well in classifying both categorical and continuous outcome variables. The algorithm divides the sample into two or more homogeneous sets based on the most significant features/predictor variables [18].

We used various feature selection techniques such as linear support vector classifier (SVC), Lasso, chi-2, Recursive Feature Elimination (RFE) and Logistic Regression, Recursive Feature Elimination (RFE) and Random Forest Classifier for selecting features in this study. Intersections of features generated by the different techniques were included for further prediction analysis. For feature selection with random forest classifier we build model with default parameter of *10* decision trees. Then we build the model by increasing the estimator to 100 decision trees. The model accuracy score with ten decision trees was 0.6055 and 0.7565 with 100 decision trees. For ML, we randomly split the un-weighted sample of 7795 (4,691 with anaemia, 3,104 without anaemia) into two: we randomly selected 80% of 7795 (n = 6235) for the training dataset and 20% (n = 1559) for the test dataset. The training dataset has been used to train the machine and the performance of the model in predicting the outcome variable was evaluated against the actual outcome variable using the test dataset, and the validation dataset was considered for the parameter estimates to be incorporated in the training models. The synthetic minority over-sampling technique (SMOTE) method was used to handle the class imbalance [30].

We evaluated the predictive performance of the models using two parameters: *calibration* and *discrimination* [31]. We used calibration plots to measure the level of agreement between the predicted probabilities of the presence of anaemia using each model versus the observed frequencies of anaemia. The discrimination of the models was assessed using the area under the Curve (AUC) estimated using the receiver characteristic curve (ROC). Sensitivity, specificity, accuracy, negative predictive value (NPV), and positive predictive value (PPV) were also estimated for each model. These evaluation metrics were calculated using the observed childhood anaemia as the reference standard. Finally, we identified the most important predictors of childhood anaemia based on the most accurate algorithms.

## Results

### Childhood anaemia by background characteristics

Of 8482 children (4882[57.6%, 95% CI: 55.0–60.1]) had anaemia. Table 1 shows the prevalence of childhood anaemia across different maternal and children's background characteristics. The proportion of children with anaemia was higher among children aged between 6–23 months (72.0%) as compared to children aged between 24–59 months (50.1%), 71.9% of children born from mother age category < 20 years and 50.2% of children borne from old mother (> = 40 years of age) had anaemia, respectively. Anaemia was relatively higher in rural areas (58.5%) than in urban areas (49.6%). The prevalence of anaemia was also higher among children from the poorest household (68.1%) versus 48.3% of children from the richest household; children with morbidity had anaemia (61%), while 56.4% of children without morbidity had anaemia.

### Predicting childhood anaemia

Table 2 shows performance of the ML algorithms based on training data. The four ML algorithms generated the predicted childhood anaemia after the trained ML-based dataset (Table 3). The predictive performance of the algorithms was evaluated using an evaluation matrix called the confusion matrix. All the algorithms gave almost the same level of accuracy. The confusion matrix shows that the logistic regression model predicted correctly 775 children had anaemia while 260 children had no anaemia. It has wrongly predicted 361 children had anaemia and 163 children without anaemia. The logistic regression model is relatively better than the rest of the models in predicting childhood anaemia with classification accuracy of 66% (95% CI: 64.0–68.7), sensitivity of 82% (95% ci: 80.2–85.0), specificity of 42% (95% CI: 38.0–45.7), and an AUC of 0.69 (95% CI: 0.66–0.72) in detecting child with anaemia. The RF algorithm was the second-best algorithm in predicting a child with anaemia. Among all algorithms used, the decision tree gave the least accuracy.

Moreover, a visualization of the receiver operating characteristics (ROC) curve was shown in Fig 1 for each algorithm. Among the four machine learning models used in this study, the logistic regression model shows relatively the highest AUC value, indicating the best in discriminating children with and without anaemia.

We also estimated and compared the accuracy of the four models using the calibration plot (Fig 2). The calibration plot measures the level of agreement between the average predicted probabilities of anaemia predicted by the models (X-axis) versus the observed anaemia frequency (Y-axis). The plot shows that the calibration by logistic regression is good, as the mean predicted probability of childhood anaemia is similar to the observed childhood anaemia across the entire distribution. The prediction from the other models is far from the observed frequency. The Hosmer-Lemeshow goodness of fit test had a P value of 0.252, indicating that the model does not misrepresent the data.

Figs 3 and 4 show the variable importance measures based on the logistic regression and RF models. Both models gave high importance to the poorest family, children aged between 6 and 23 months, unemployed mothers, mothers with no formal education, and child stunting, and are among the top seven predictors of childhood anaemia. Maternal anaemia and mother's age < 20 are given high importance in logistic regression but the least in the RF algorithm.

## Discussion

We attempted to explore; the 2016 Ethiopian demographic health survey dataset to provide an initial insight into the potential applicability of machine learning methods in predicting childhood anaemia status based on socio-demographic and health characteristics. A logistic

**Table 1. Childhood anaemia status by maternal and child characteristics.**

| Variables | Anaemia (%), n = 4882 | No anaemia (%), n = 3600 | Chi-square test of equality |
|---|---|---|---|
| Child sex | | | P = 0.934 |
| Male | 2534 (57.6) | 1864 (42.4) | |
| Female | 2348 (57.5) | 1736 (42.5) | |
| Child age in months | | | P < 0.001 |
| 6 −23 | 2091 (72.0) | 814 (28.0) | |
| 24 −59 | 2791 (50.1) | 2786 (49.9) | |
| Age of mother | | | P = 0.002 |
| < 20 | 162 (71.9) | 63 (28.1) | |
| 20−29 | 2457 (59.4) | 1680 (40.6) | |
| 30−39 | 1869 (56.1) | 1466 (44.0) | |
| > = 40 | 394 (50.2) | 391 (49.8) | |
| Place of residence | | | P = 0.005 |
| Rural | 4455 (58.5) | 3166 (41.5) | |
| Urban | 427 (49.6) | 434 (50.4) | |
| Wealth quintile | | | P < 0.001 |
| Poorest | 1354 (68.1) | 634 (31.9) | |
| Poorer | 1151 (57.9) | 838 (42.1) | |
| Middle | 980 (53.8) | 843 (46.2) | |
| Richer | 845 (54.8) | 695 (45.2) | |
| Richest | 552 (48.3) | 590 (51.7) | |
| Mother's education | | | P = 0.03 |
| No education | 3332 (58.6) | 2353 (41.4) | |
| Primary education | 1298 (56.9) | 983 (43.1) | |
| Secondary education | 168 (48.7) | 177 (51.3) | |
| Higher education | 84 (49.1) | 87 (50.9) | |
| Employment status of mother | | | P < 0.001 |
| Unemployed | 2839 (61.4) | 1787 (38.6) | |
| Employed | 2043 (53.0) | 1813 (47.0) | |
| Source of drinking water | | | P < 0.001 |
| Non- improved | 4566 (58.4) | 3247 (41.6) | |
| Improved | 316 (47.3) | 353 (52.7) | |
| Drug for parasite within 6 months | | | P = 0.240 |
| Yes | 599 (54.9) | 492 (45.1) | |
| No | 4283 (58.0) | 3108 (42.0) | |
| Stunting | | | P = 0.001 |
| Yes | 1876 (61.9) | 1155 (38.1) | |
| No | 3006 (55.2) | 2445 (44.8) | |
| Wasting | | | P = 0.001 |
| Yes | 520 (67.5) | 250 (32.5) | |
| No | 4363 (56.6) | 3350 (43.4) | |
| Child morbidity | | | P = 0.022 |
| Yes | 1271 (61.0) | 814 (39.0) | |
| No | 3611 (56.4) | 2786 (43.6) | |
| Maternal anaemia | | | P = 0.024 |
| Yes | 95 (74.2) | 33 (25.8) | |
| No | 4788 (57.3) | 3567 (42.7) | |

**Table 2. Performance indicators of the four ML algorithms as evaluated on training data.**

| Evaluation parameters | | Logistic regression | | | Decision tree | | | Random forest | | | KNN | | |
|---|---|---|---|---|---|---|---|---|---|---|---|---|---|
| Confusion matrix | Observed | Predicted | | | Predicted | | | Predicted | | | Predicted | | |
| | | No anaemia | No anaemia | Anaemia | No anaemia | No anaemia | Anaemia | No anaemia | No anaemia | Anaemia | No anaemia | No anaemia | Anaemia |
| | | | 758 | 1104 | | 892 | 970 | | 805 | 1057 | | 858 | 1004 |
| | | Anaemia | 587 | 2228 | Anaemia | 510 | 2305 | Anaemia | 423 | 2392 | Anaemia | 581 | 2234 |
| | | % (95% CI) | | | % (95% CI) | | | % (95% CI) | | | % (95% CI) | | |
| Accuracy | | 64 (62.5–65.2) | | | 68 (67.0–69.7) | | | 68 (67.0–69.7) | | | 66 (64.7–67.5) | | |
| Sensitivity | | 79 (77.6–80.6) | | | 82 (80.4–83.3) | | | 85 (83.7–86.3) | | | 79 (77.9–80.9) | | |
| Specificity | | 41 (38.5–42.9) | | | 48 (45.6–50.2) | | | 43 (41.0–45.5) | | | 46 (43.8–48.3) | | |
| Positive predictive value | | 67 (65.3–68.4) | | | 70 (68.8–71.9) | | | 69 (67.8–70.9) | | | 69 (67.4–70.6) | | |
| Negative predictive value | | 56.4(53.7–59.0) | | | 64 (61.1–66.1) | | | 66 (62.9–68.2) | | | 59 (57.1–62.1) | | |
| AUC | | 66 (64.4–67.6) | | | 74 (72.5–75.5) | | | 74 (72.5–75.5) | | | 69 (67.4–70.6) | | |
| Performance time | | 0.01 seconds | | | 0.01 seconds | | | 0.45 seconds | | | 0.1 seconds | | |

regression algorithm shows a higher predictive performance compared other ML algorithms used in this study; with this regard to this, the logistic regression algorithm shows that most of the variables related to maternal and child health such as maternal age category < 20, a mother with no formal education, household wealth index poorest category, unemployed mother, maternal anaemia, age of child, child nutritional status (stunting, wasting), and child morbidity are predictors of childhood anaemia in Ethiopia. Likewise, the RF also gave almost similar results as logistic regression except for the level of importance.

The logistic regression algorithm shows an accuracy of 66% (95% CI: 62.3–68%), which is similar to the study conducted in Bangladesh shows an accuracy of 63% (95% CI: 58.4–70%) in predicting childhood anaemia however, the RF shows the best performance compared to other ML algorithms in studies from Bangladesh [23] Afghanistan [32], to the contrary study by Appiahene P et al, shows that Naïve Bayes model performs better (accuracy of 99%) compared to K-NN and Support vector machine models [33].

Several variables contributed to the prediction of childhood anaemia. However, the level of importance varies between the logistic regression and RF models. The logistic regression model revealed that a child aged between 6 and 23 months, mother's age < 20 years old,

**Table 3. Performance indicators of the four ML algorithms as evaluated on test data.**

| Evaluation parameters | | Logistic regression | | | Decision tree | | | Random forest | | | KNN | | |
|---|---|---|---|---|---|---|---|---|---|---|---|---|---|
| Confusion matrix | Observed | Predicted | | | Predicted | | | Predicted | | | Predicted | | |
| | | No anaemia | No anaemia | Anaemia | No anaemia | No anaemia | Anaemia | No anaemia | No anaemia | Anaemia | No anaemia | No anaemia | Anaemia |
| | | | 260 | 361 | | 286 | 335 | | 259 | 362 | | 236 | 385 |
| | | Anaemia | 163 | 775 | Anaemia | 281 | 657 | Anaemia | 225 | 713 | Anaemia | 199 | 739 |
| | | % (95% CI) | | | % (95% CI) | | | % (95% CI) | | | % (95% CI) | | |
| Accuracy | | 66 (64.0–68.7) | | | 60 (58.0–62.9) | | | 64 (60.0–65.8) | | | 62 (60.1–64.0) | | |
| Sensitivity | | 82 (80.2–85.0) | | | 70 (67.1–73.0) | | | 79 (76.2–81.4) | | | 76 (73.3–78.7) | | |
| Specificity | | 42 (38.0–45.7) | | | 46 (42.1–50.0) | | | 42 (37.8–45.6) | | | 38 (34.2–45.8) | | |
| Positive predictive value | | 68 (65.4–70.9) | | | 66 (63.2–69.2) | | | 66 (63.4–69.2) | | | 66 (62.9–68.5) | | |
| Negative predictive value | | 61 (56.6–66.1) | | | 50 (46.2–54.6) | | | 54 (49.0–58.0) | | | 54 (49.4–59.0) | | |
| AUC | | 69 (66.4–71.6) | | | 60 (57.2–62.8) | | | 63 (60.2–65.8) | | | 62 (61.3–66.7) | | |
| Performance time | | 0.02 seconds | | | 0.02 seconds | | | 0.45 seconds | | | 0.1 seconds | | |

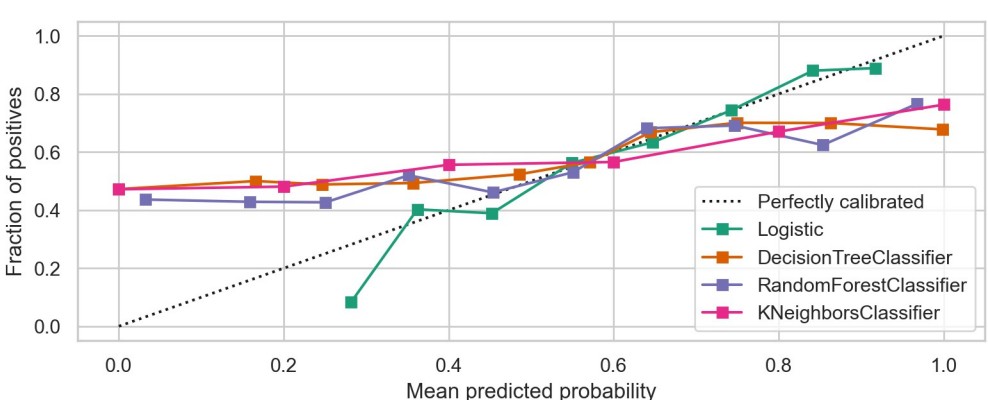

**Fig 1. ROC Curves for the four models.**

**Fig 2. Calibration plots for the four models.**

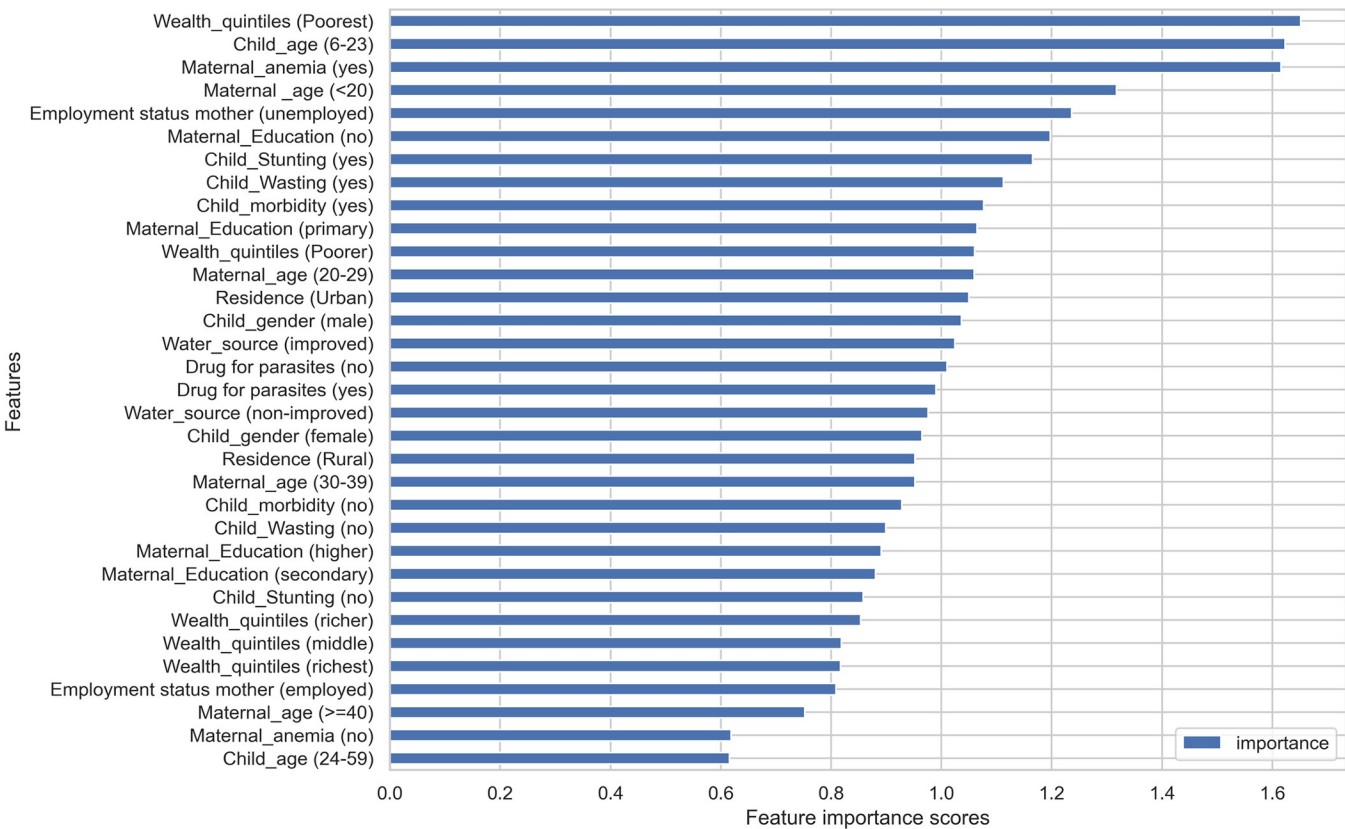

**Fig 3. Variable importance measures for the logistic regression algorithm.**

mother's anaemia, child stunting, and wasting are the most important predictors of childhood anaemia. This finding is consistent with a study from India [24]. Child nutrition, mother's age, unemployed mother, no drug for parasites within the previous 6 months, and child morbidity are important predictors identified by both logistic regression and RF consistently, even though their degree of importance varies. However, the two algorithms identified inconsistent predictors like child gender (male) given high importance in logistic regression, while child gender (female) was given high importance in the RF algorithm. Moreover, maternal anaemia was identified as the second top important predictor of childhood anaemia in logistic regression but was the least in RF.

The high importance scores given to the poorest wealth quintile, by both logistic regression and RF, may be explained by the fact that children with low-income families are at high risk of iron deficiency and more prone to develop iron deficiency anaemia as evidence shows from a study conducted among healthy children in Canada [34]. Moreover, another study from the United States shows that children from the poorest families are at high risk of food insecurity, and children from such families are more likely to have iron deficiency anaemia [35]. Another interesting finding of this study is that both logistic regression and the RF models gave high importance to mothers with no education and unemployed mothers. These two predictors are proxies for poverty. A Nigerian study conducted among children with sickle cell anaemia suggested that severe anaemia is associated with a lower educational level of household [36].

The RF model shows high importance to employed mothers, but the logistic regression gives relatively the lowest score. A study from India reveals that if both couples have jobs, the

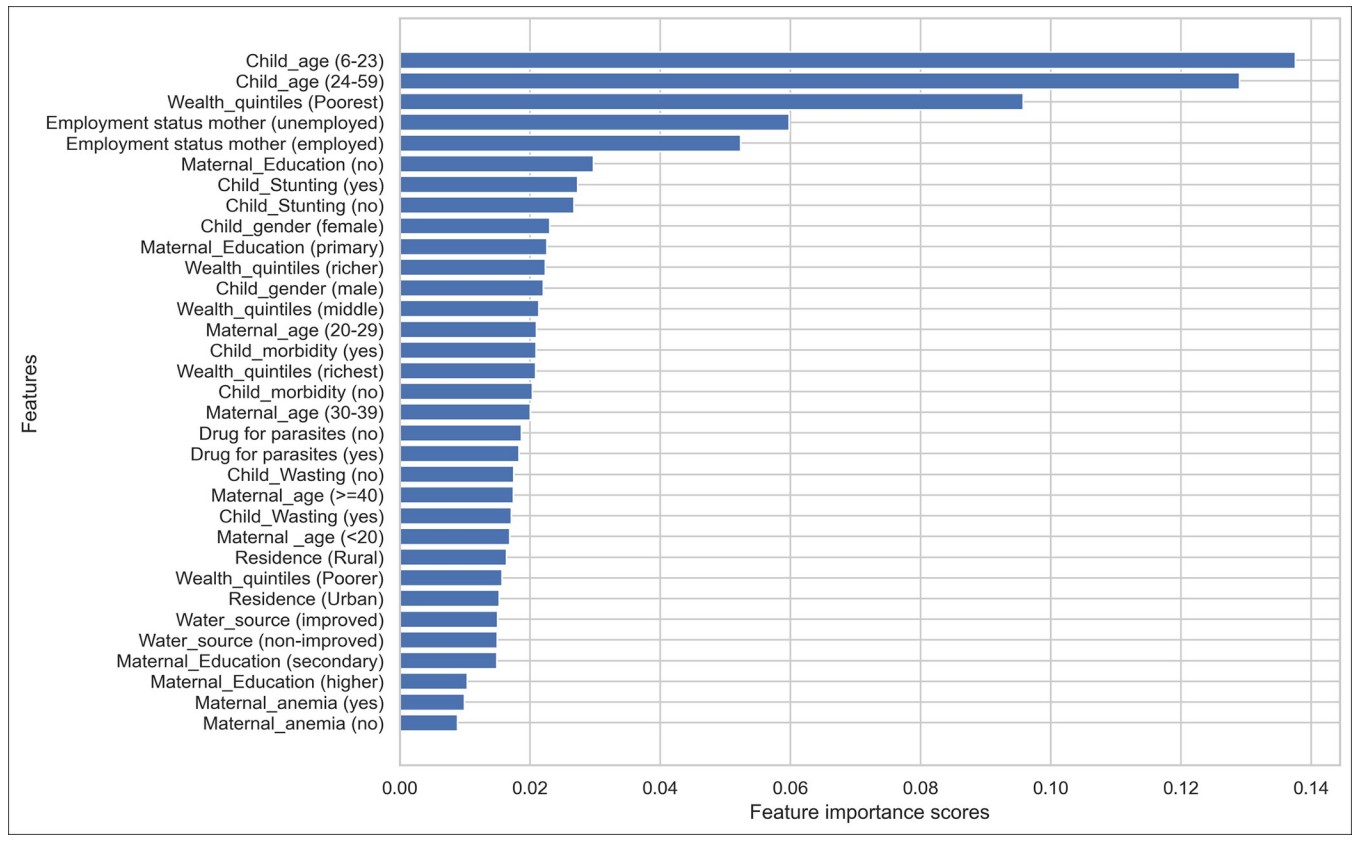

**Fig 4. Variable importance for the random forest algorithm.**

risk of having anaemia is significantly high for the child [37]. Uneducated and unemployed mothers and poor households might be unable to afford and have access to adequate nutrients rich in iron and other micronutrient-rich food.

A study using data from four African countries (Malawi, Mozambique, Namibia, and Zimbabwe) using mixed models shows that maternal anaemia is a predictor of childhood anaemia in all those four countries [38]. According to a study from India, iron supplementation during pregnancy has effects on a child's haemoglobin level [39]. Similarly, our ML-based analysis shows that maternal anaemia is identified as the top predictor in logistic regression even though the RF gave the least scores.

Younger children aged 6 to 23 months were predictors of anaemia compared to older children. This finding is supported by a study from Malawi [40] and China [41]. There is an increasing need for iron and micronutrient-rich food in the first year of life for growth and development [42]. However, due to different reasons such as poverty and low education, mothers or caretakers cannot afford to provide adequate nutrients and iron-reach food for their children.

We define child morbidity as having fever, diarrhoea, or acute respiratory infections. The logistic regression and RF ML algorithms identified as important predictors of childhood anaemia. Previous studies using advanced statistical techniques show that children with fever are at high risk of having anaemia [40].

Endris BS and his colleagues from Ethiopia use the Bayesian Geospatial statistical modelling of data from the 2016 EDHS shows child wasting, stunting, maternal anaemia, and child from

poorer family are independent predictors of childhood anaemia [28]. However, our analysis of similar data using ML algorithms identified other predictors of childhood anaemia that are were not identified by the Bayesian Geospatial statistical modelling such as child age between 6 and 23 months, childhood morbidity, maternal education, and maternal age.

The strength of this study is that it uses nationally representative data that assures its generalizability and applicability by primary healthcare professionals. Moreover, we utilized different ML algorithms that help to catch a combination of predictors of childhood anaemia, which would have remained undetected; if a single statistical analysis technique had been used. However, the study is not without limitations: First, since we used secondary data we lack some clinical features of anaemia such as pallor of the palm, conjunctiva, and tongue which would improve the moderate performance of the ML algorithms observed in this study. Therefore application of ML algorithms to predict childhood anaemia using these clinical important variables can be a potential research area for the future. Second, data such as fever, diarrhoea, and cough in the past two weeks preceding the survey relied on mothers' interviews which are prone to recall bias. Lastly, we didn't develop clinical scores (risk score) and their corresponding probability for each identified predictors which, would have been more practical for healthcare professionals to use in their day-to-day clinical practice.

## Conclusions

The findings show that the logistic regression ML algorithm offers better prediction accuracy than the other ML algorithms used in this study, followed by the RF algorithm. Furthermore, the logistic regression and the RF ML algorithms have shown diverse combinations of the most important predictors of childhood anaemia, though degree of importance varying between the two ML algorithms. Therefore, the study provides evidence on how the ML-based approach can be used to better understand predictors of childhood anaemia in addition to the traditional statistical analysis approach used. The findings might be implemented by the health care practitioners in the primary health care unit in their daily clinical practice working at rural primary health care units where the diagnostic facility to identify anaemia is poor.

## Acknowledgments

We would like to thank the demographic health survey programme for providing free access to the data set used for this analysis.

## Author Contributions

**Conceptualization:** Solomon Hailemariam Tesfaye.

**Formal analysis:** Solomon Hailemariam Tesfaye.

**Software:** Solomon Hailemariam Tesfaye.

**Writing – original draft:** Solomon Hailemariam Tesfaye.

**Writing – review & editing:** Solomon Hailemariam Tesfaye, Binyam Tariku Seboka, Daniel Sisay.

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
