## [Decision Letter · Decision Letter 0]

11 Dec 2023

PONE-D-23-22400Application of machine learning methods for predicting childhood anaemia: Analysis of Ethiopian Demographic Health Survey of 2016.PLOS ONE

Dear Dr. Tesfaye,

Thank you for submitting your manuscript to PLOS ONE. After careful consideration, we feel that it has merit but does not fully meet PLOS ONE’s publication criteria as it currently stands. Therefore, we invite you to submit a revised version of the manuscript that addresses the points raised during the review process.

We look forward to receiving your revised manuscript.

Kind regards,

Mohammed Moinuddin, PhD

Academic Editor

PLOS ONE

Journal Requirements:

Additional Editor Comments:

Please do not forget to focus on the feedbacks and suggestions provided by our reviewers.

Reviewers' comments:

Reviewer's Responses to Questions

**Comments to the Author**

1. Is the manuscript technically sound, and do the data support the conclusions?

Reviewer #1: Yes

Reviewer #2: Yes

2. Has the statistical analysis been performed appropriately and rigorously? 

Reviewer #1: Yes

Reviewer #2: Yes

3. Have the authors made all data underlying the findings in their manuscript fully available?

Reviewer #1: Yes

Reviewer #2: Yes

4. Is the manuscript presented in an intelligible fashion and written in standard English?

Reviewer #1: Yes

Reviewer #2: Yes

5. Review Comments to the Author

Reviewer #1: 1. It is unclear why specific ML techniques were chosen for this study. Could you please provide a detailed explanation? Additionally, it would be beneficial to understand the rationale behind opting for ML techniques over traditional statistical tools.

2. My understanding is that accuracy was considered for measuring the performance of the three algorithms. I recommend including performance time as an additional metric to provide a more comprehensive evaluation.

3. To enhance clarity, consider incorporating a table in the introduction section. This table could highlight the research gap in existing articles and compare it with the proposed research, offering a visual aid for readers.

4. In the section on study variables and measurements, it's mentioned that samples were taken from heel pricks for ages 6-11 months. Could you elaborate on why this age range was chosen and why other age groups were not included?

5. It's not clear whether statistical tools were employed to assess the association of selected variables. Providing details on the statistical analyses conducted would enhance the clarity of this aspect.

6. Please elaborate on how household wealth quintiles are classified and defined. Including this information in the methodology section would enhance the overall understanding of the study.

7. Kindly review Table 1 and the write-up in the results section to address any discrepancies in the values provided. Consistency between the table and the accompanying text is crucial for accurate interpretation.

8. Consider adding a section on the potential contribution of the article to the scientific community. Additionally, sharing insights into the challenges faced during the study would provide valuable context for readers.

Reviewer #2: ### Overview of the Paper

The paper applies machine learning (ML) techniques to predict childhood anemia using the 2016 Ethiopian Demographic Health Survey (EDHS) data. It explores various socio-demographic, economic, and maternal and child health variables to improve anemia diagnosis in resource-poor settings. While the application of ML in healthcare is not new, the use of recent datasets, such as the 2021 EDHS in addition to the 2016 EDHS, provides valuable insights for practical healthcare applications in Ethiopia. I recommend adding the results with recent data, at least briefly, to provide insight into practical implications.

### Review Comments

1. Technical Soundness: The manuscript is technically sound, with a well-justified methodology, appropriate statistical analysis, and data-supported conclusions. The data used has been made available, and the manuscript is structured effectively.

2. Novelty and Validation: Although ML methods in healthcare are established, their application to recent datasets, such as the 2021 EDHS, in addition to the 2016 EDHS, is beneficial. However, further validation in different settings with newer data would strengthen the findings.

3. Limitations and Practical Implications: The authors should more explicitly address the study's limitations, especially regarding the moderate predictive power of the ML model and the limitations inherent in the EDHS data. Additionally, elaborating on the practical implementation of these findings in Ethiopian healthcare settings would enhance the paper's value.

4. Abstract: The abstract requires rewriting to eliminate repetition, particularly in the second and third lines.

5. Inconsistency in Sample Size: There is a discrepancy in the reported sample sizes (8482 vs 7795) in the data and statistical analysis sections. This discrepancy needs clarification.

6. Comparative Methods and Parameters: The paper should include a brief discussion on the comparative methods used, with references for further details in the Methods section. Details on the selection of optimum parameters, like the optimum number of trees in the Random Forest model, which may affect the predictive power, are also needed.

7. Missing Values and Model Accuracy: Information about the treatment of missing values is lacking. Clarification on whether complete cases were used in the analysis is necessary. Additionally, model accuracy metrics based on training data should be provided, similar to the test data metrics in Table 2.

6. PLOS authors have the option to publish the peer review history of their article (what does this mean?). If published, this will include your full peer review and any attached files.

Reviewer #1: **Yes: **Md. Murad Hossain

Reviewer #2: **Yes: **Dr. Md Jamal Hossain

---

## [Author Response · Author response to Decision Letter 0]

28 Dec 2023

Date: December 28, 2023

Dr Mohammed Moinuddin

Academic editor

PLOS ONE

 “Application of machine learning methods for predicting childhood anaemia: Analysis of Ethiopian Demographic Health Survey of 2016."

Dear Editor 

Thank you for the opportunity to submit our revised manuscript “Application of machine learning methods for predicting childhood anaemia: Analysis of Ethiopian Demographic Health Survey of 2016" for the consideration for publication in PLOS ONE journal. We are grateful for the critical and valuable comments. Below we provide a point-by-point response to each of the reviewers. All the changes made are indicated with track changes in the file labelled “Revised manuscript with Track Changes.” We also address the journal requirements raised by the Editor such as data availability statement. 

We appreciate your further consideration of this manuscript.

Solomon Hailemariam Tesfaye (PhD)

Corresponding author

Below is our point-by-point response to the comments given by reviewers. 

Responses:

Reviewer 1 comments:

Comment: It is unclear why specific ML techniques were chosen for this study. Could you please provide a detailed explanation? Additionally, it would be beneficial to understand the rationale behind opting for ML techniques over traditional statistical tools.

Response: ML techniques focused on making prediction as accurate as possible, while the traditional statistical methods are aimed at inferring relationship between variables. Unlike the traditional statistical methods machine learning algorithms are data driven and free from prior assumptions, but the traditional statistical methods are model driven. ML shows improvement over the traditional methods in predicting diseases in health care fields. The diagnosis of childhood anaemia is resource intensive and challenging particularly in rural settings where there is scarcity of resources. Previous studies show improved ML performance in predicting childhood anaemia. Therefore, the aim of this study is to apply a machine learning (ML) algorithm to predict childhood anaemia using socio-demographic, economic, and maternal and child-related variables. We believe that the findings from this analysis will help healthcare providers to easily identify and treat/refer children with anaemia. We have added this paragraph in the revised manuscript (page 3, lines 56-66). 

Comment: My understanding is that accuracy was considered for measuring the performance of the three algorithms. I recommend including performance time as an additional metric to provide a more comprehensive evaluation.

Response: We provided the performance time for each of the ML algorithms (see pages 11 and 12, tables 2 and 3). 

Comment: 3. To enhance clarity, consider incorporating a table in the introduction section. This table could highlight the research gap in existing articles and compare it with the proposed research, offering a visual aid for readers.

Response: We accept the comment and add the following message in the box in the revised manuscript (see page 3, lines: 68-74). 

Comment: In the section on study variables and measurements, it's mentioned that samples were taken from heel pricks for ages 6-11 months. Could you elaborate on why this age range was chosen and why other age groups were not included?

Response: According to Ethiopian Demographic Health survey (EDHS) children age between 6 and 59 months are target group for anaemia measurement. For children aged 12-59 months blood samples were taken from a finger prick and heel prick for children aged 6-11 months. We wrote these sentences in the manuscript (page 5, lines 99-102). Therefore no children were excluded from the target group. 

Comment: It's not clear whether statistical tools were employed to assess the association of selected variables. Providing details on the statistical analyses conducted would enhance the clarity of this aspect.

Response: As we have mentioned in the manuscript (page 5, lines 90-100) thirteen variables (features) were selected based on their clinical relevance and their association with anaemia using findings from nine previous studies conducted in Ethiopia. We believe it would be duplication to conduct the statistical analysis to see and select significantly associated variables. Moreover, among the nine previous studies mentioned, one of the studies used the same data set we used (EDHS-2016) to identify risk factors of childhood anaemia using the traditional statistical methods. Our study aims at building several predictive models using the already established risk factors of anaemia in children through ML approach based on the Ethiopian Demographic and Health Survey (EDHS) data (see page 5, lines 108-117). We also used various feature selection techniques before training the machine (see page 6-7, lines: 143-150). 

Comment: Please elaborate on how household wealth quintiles are classified and defined. Including this information in the methodology section would enhance the overall understanding of the study.

Response: The Ethiopian demographic health survey collect data from households ranging from a television to a bicycle or car possession, in addition to housing characteristics such as source of drinking water, toilet facilities, and flooring materials. Households are given scores based on the number and kinds of consumer goods they own. These scores are derived using principal component analysis. National wealth quintiles are compiled by assigning the household score to each household. We added this information in the revised manuscript (pages 5-6, lines: 118-123). 

Comment: Kindly review Table 1 and the write-up in the results section to address any discrepancies in the values provided. Consistency between the table and the accompanying text is crucial for accurate interpretation.

Response: We accept the comment and revised the values and the texts (see page 8, lines: 174-180). 

Comment: Consider adding a section on the potential contribution of the article to the scientific community. Additionally, sharing insights into the challenges faced during the study would provide valuable context for readers.

Response: The study didn’t collect data on some of the very important clinical symptoms of anaemia such as pallor of the palm, conjunctiva, and tongue. Therefore application of ML algorithms using these clinical important variables to diagnose childhood anaemia can be a potential research area for the future. This sentence is now added in the revised manuscript (see page 16, lines: 293-298). 

Reviewer 2 comments: 

Comment: Overview of the Paper

The paper applies machine learning (ML) techniques to predict childhood anaemia using the 2016 Ethiopian Demographic Health Survey (EDHS) data. It explores various socio-demographic, economic, and maternal and child health variables to improve anaemia diagnosis in resource-poor settings. While the application of ML in healthcare is not new, the use of recent datasets, such as the 2021 EDHS in addition to the 2016 EDHS, provides valuable insights for practical healthcare applications in Ethiopia. I recommend adding the results with recent data, at least briefly, to provide insight into practical implications.

Response: As commented by the reviewer it would have been better if the analysis was based on the recent data of EDHS. Our plan was to use the 2019/2021 EDHS data. But, by the time we start writing up the protocol either the 2019 or 2021 EDHS was not released. At time of writing this response to the reviewers the 2021 EDHS data was not yet released and only preliminary data of the 2019 EDHS was released. The 2019 EDHS preliminary is not full dataset and childhood anaemia is not available. 

Comment: Technical Soundness: The manuscript is technically sound, with a well-justified methodology, appropriate statistical analysis, and data-supported conclusions. The data used has been made available, and the manuscript is structured effectively.

Comment: Novelty and Validation: Although ML methods in healthcare are established, their application to recent datasets, such as the 2021 EDHS, in addition to the 2016 EDHS, is beneficial. However, further validation in different settings with newer data would strengthen the findings.

Response: We accept the comment; however because of the unavailability of the 2021 data we are unable to add results from this recent datasets. Validation of the model performance can be a potential research area for the future. 

Comment: Limitations and Practical Implications: The authors should more explicitly address the study's limitations, especially regarding the moderate predictive power of the ML model and the limitations inherent in the EDHS data. Additionally, elaborating on the practical implementation of these findings in Ethiopian healthcare settings would enhance the paper's value.

Response: Since we use secondary data we lack some clinical features of anaemia such as pallor of the palm, conjunctiva, and tongue which would improve the moderate performance of the ML algorithms observed in this study (see page 16, lines: 293-298). Since the other reviewer of this study also asked the potential contribution/implication of this study, we add sentences related with the implication of study in the revised manuscript (see page 3, sentences in the box). 

Comment: Abstract: The abstract requires rewriting to eliminate repetition, particularly in the second and third lines.

Response: According to the comment we amend the sentences (see page 1, lines: 9-11). 

Comment: Inconsistency in Sample Size: There is a discrepancy in the reported sample sizes (8482 vs 7795) in the data and statistical analysis sections. This discrepancy needs clarification.

Response: The numbers of children with complete hemoglobin measurement was 7795 and the ML analysis was based on 7795 (see page 7, lines: 150-152). But as mentioned in the manuscript (page 4, lines: 95-97) we use sampling weight (8442) to estimate the prevalence of anaemia (as recommended by the Demographic health programme guideline) to restore the representativeness of the data and to get more accurate results. Therefore the sample (8442) mentioned in the method section is the weighted sample for descriptive results.

Comment: Comparative Methods and Parameters: The paper should include a brief discussion on the comparative methods used, with references for further details in the Methods section. Details on the selection of optimum parameters, like the optimum number of trees in the Random Forest model, which may affect the predictive power, are also needed.

Response: We accept the comment and we provide detail discussion about the four ML algorithm used in the analysis (page 6, lines: 132-142). According to recommendation by Geron A, we used various feature selection techniques such as linear support vector classifier (SVC), Lasso, chi-2, Recursive Feature Elimination (RFE) and Logistic Regression, Recursive Feature Elimination (RFE) and Random Forest Classifier for selecting features in this study (pages 6, lines: 143-146). Intersections of features generated by the different techniques were included for further prediction analysis. For feature selection with random forest classifier we build model with default parameter of estimator=10. Then we build the model by increasing the estimator to 100 decision trees. The model accuracy score with 10 decision trees was 0.6055 and 0.7565 with 100 decision trees. As expected the model accuracy increases with increasing decision trees. We then use the feature selection with the 100 decision trees. This detail information is provided in the revised manuscript (see pages 6-7, lines: 146-150).

 Comment: Missing Values and Model Accuracy: Information about the treatment of missing values is lacking. Clarification on whether complete cases were used in the analysis is necessary. Additionally, model accuracy metrics based on training data should be provided, similar to the test data metrics in Table 2.

Response: Cases with multiple missing values were excluded from analysis. We used only cases with complete information. Accordingly we provided this information in the revised manuscript (page 4, lines: 94-95). We add model performance matrix based on training data and the table is now available in the revised manuscript (see page 11, table 2).

---

## [Decision Letter · Decision Letter 1]

23 Feb 2024

Application of machine learning methods for predicting childhood anaemia: Analysis of Ethiopian Demographic Health Survey of 2016.

PONE-D-23-22400R1

Dear Dr. Tesfaye,

We’re pleased to inform you that your manuscript has been judged scientifically suitable for publication and will be formally accepted for publication once it meets all outstanding technical requirements.

Kind regards,

Mohammed Moinuddin, PhD

Academic Editor

PLOS ONE

Additional Editor Comments (optional):

Reviewers' comments:

Reviewer's Responses to Questions

**Comments to the Author**

1. If the authors have adequately addressed your comments raised in a previous round of review and you feel that this manuscript is now acceptable for publication, you may indicate that here to bypass the “Comments to the Author” section, enter your conflict of interest statement in the “Confidential to Editor” section, and submit your "Accept" recommendation.

Reviewer #1: All comments have been addressed

Reviewer #2: All comments have been addressed

2. Is the manuscript technically sound, and do the data support the conclusions?

Reviewer #1: Yes

Reviewer #2: Yes

3. Has the statistical analysis been performed appropriately and rigorously? 

Reviewer #1: I Don't Know

Reviewer #2: Yes

4. Have the authors made all data underlying the findings in their manuscript fully available?

Reviewer #1: Yes

Reviewer #2: Yes

5. Is the manuscript presented in an intelligible fashion and written in standard English?

Reviewer #1: Yes

Reviewer #2: Yes

6. Review Comments to the Author

Reviewer #1: Thank you all authors for your nice clarification and updating your articles according to my given comments.

Reviewer #2: No additional comments. All the suggestions and comments provided by the reviewers have been addressed and, if possible, satisfied. Recommended for publication.

7. PLOS authors have the option to publish the peer review history of their article (what does this mean?). If published, this will include your full peer review and any attached files.

Reviewer #1: **Yes: **Md. Murad Hossain

Reviewer #2: **Yes: **Dr MD Jamal Hossain

---

## [Editor Report · Acceptance letter]

27 Feb 2024

PONE-D-23-22400R1 

PLOS ONE

Dear Dr. Tesfaye, 

I'm pleased to inform you that your manuscript has been deemed suitable for publication in PLOS ONE. Congratulations! Your manuscript is now being handed over to our production team.

Kind regards, 

on behalf of

Dr Mohammed Moinuddin 

Academic Editor

PLOS ONE